# Complete Abdominal Evisceration After Open Hysterectomy: A Case Report and Evidence-Based Review

**DOI:** 10.3390/jcm14010262

**Published:** 2025-01-05

**Authors:** Valentin Nicolae Varlas, Irina Bălescu, Roxana Georgiana Varlas, Al-Aloul Adnan, Alexandru George Filipescu, Nicolae Bacalbașa, Nicolae Suciu

**Affiliations:** 1Department of Obstetrics and Gynecology, Filantropia Clinical Hospital, 011132 Bucharest, Romania; valentin.varlas@umfcd.ro; 2Department of Obstetrics and Gynaecology, “Carol Davila” University of Medicine and Pharmacy, 050474 Bucharest, Romania; alexandru.filipescu@umfcd.ro; 3Department of Surgery, Ponderas Academic Hospital, 021188 Bucharest, Romania; irina_balescu206@yahoo.com; 4Ramnicu Sarat Municipal Hospital, 125300 Buzau, Romania; adnanalaloul@yahoo.com; 5Faculty of Nursing, Bioterra University, 013724 Bucharest, Romania; 6Department of Obstetrics and Gynaecology, Elias Emergency University Hospital, 011461 Bucharest, Romania; 7Department of Visceral Surgery, Fundeni Clinical Institute, 022328 Bucharest, Romania; nicolaebacalbasa@gmail.com; 8Fetal Medicine Excellence Research Center, Alessandrescu-Rusescu National Institute for Mother and Child Health, 020395 Bucharest, Romania; nicolae.suciu@umfcd.ro; 9Division of Obstetrics, Gynecology and Neonatology, Carol Davila University of Medicine and Pharmacy, 050474 Bucharest, Romania; 10Department of Obstetrics and Gynecology, Alessandrescu-Rusescu National Institute for Mother and Child Health, Polizu Clinical Hospital, 020395 Bucharest, Romania

**Keywords:** dehiscence of the abdominal surgical wound, abdominal evisceration, open abdomen, burst abdomen, abdominal fascial dehiscence, complications of hysterectomy

## Abstract

**Background/Objectives:** Despite its low incidence, complete postoperative abdominal evisceration represents a complication requiring an urgent solution. We aimed to present a rare case of an abdominal evisceration of the omentum and small-bowel loops after a total abdominal hysterectomy and review the literature regarding this condition’s diagnosis and therapeutic management. **Case report:** On the sixth postoperative day for a uterine fibroid, a 68-year-old patient presented with an abdominal evisceration of the omentum and small bowel that occurred two hours before. An emergency laparotomy was performed to correct the evisceration and restore the integrity of the abdominal wall structure. The literature review was carried out in the PubMed, Embase, and Web of Science databases using the terms “abdominal wall dehiscence”, “abdominal evisceration”, “open abdomen”, “burst abdomen”, “abdominal fascial dehiscence”, “abdominal dehiscence post-hysterectomy”, and “hysterectomy complications” by identifying all-time articles published in English. **Results:** Seven studies were included in this electronic search. The early diagnosis of abdominal evisceration, the identification of risk factors and comorbidities, followed by the choice of surgical technique, and postoperative follow-up were parts of the standard algorithm for managing this life-threatening case. **Conclusions:** Abdominal evisceration, as a surgical emergency, requires the diagnosis and treatment of this complication alongside the identification of the risk factors that can lead to its occurrence, as well as careful postoperative monitoring adapted to each case.

## 1. Introduction

Total abdominal hysterectomy (TAH), despite seeing a downward trend in recent years, continues to be indicated for gynecological malignancy, large fibroids, adhesion, endometriosis, chronic pelvic pain, and pelvic infection. Furthermore, a consensus has not been established regarding the hysterectomy route, which is determined by the operator’s experience after counseling the patient. Thus, the high rate of extensive surgical interventions increases the risk of postoperative complications [1]. Fortunately, frequent minimally invasive operations have decreased this risk in recent years. A 2023 survey of members of the International Society of Gynecological Endoscopy regarding current practice for the approach to hysterectomy showed that total laparoscopic hysterectomy (TLH) was preferred by 59% of members for themselves and their relatives [2].

Evisceration represents the protrusion of the intra-abdominal organs via the dehiscence of the anatomical planes of the abdominal wall. Likewise, eviscerations are divided into the following two categories: those that occur after an abdominal trauma (i.e., eviscerations that do not pose a particular problem) and eviscerations after surgical interventions. In the case of gynecological interventions, the frequent site of evisceration is the vaginal one, followed by the abdominal one [3]. The true incidence of these complications is not fully known because of ethical and legal issues. Abdominal wound dehiscence is a severe postoperative complication with an incidence between 0.4 and 3.5%, varying by gender, geographic area, and country income [4]. The evisceration process can occur postoperatively from the first days to 3 weeks, with a peak occurring on postoperative day 7 and a frequency rate of 0.3–0.7% in gynecological patients [5,6].

The most common post-hysterectomy complications described are urinary tract (bladder and ureter) and intestinal injuries, respectively, prolapse of the vaginal vault, and, much less often, eviscerations. Abdominal evisceration after hysterectomy is less frequently reported in the literature compared with vaginal cuff dehiscence. Brettauer J. was among the first to describe abdominal evisceration in 1899 [7], followed by studies on a larger number of cases by Sokolov [8] and Sedgwick over 32 years (between 1921 and 1952) [9].

Commonly implicated risk factors are catabolic changes, advanced age, obesity, cancer, peritonitis, respiratory disease, cough, constipation, previous pelvic/abdominal surgery, and radiotherapy. Eviscerations can occur following interventions in the digestive tract or pelvic organs. Thus, 96.8% of evisceration cases occur after vertical abdominal incisions (52.6%, 29.4%, and 14.7% in the upper abdomen, lower abdomen, and middle abdomen, respectively), and only 3.2% in the case of transverse or oblique incisions (1% each in the upper abdomen, subcostal, and McBurney regions). The common clinical manifestations are fever, pain, cough, abdominal distension, nausea, vomiting, changes in intestinal transit, and wound secretions or blood [10].

Abdominal evisceration of the small-bowel loops after pelvic surgery requires urgent surgical intervention. This condition, which can put the patient’s life at risk, should be managed by a multidisciplinary medical team. Thus, counseling regarding the possible complications of hysterectomy is important for patients with risk factors. Abdominal evisceration involving the omentum and small bowel is a complication with an incidence that is not fully known and is rarely reported. Without prompt surgical treatment, it can have a severe evolution, leading to further increases in morbidity and mortality in these patients.

The lack of a quick decision by the medical team can compromise the eviscerated bowel, with its viability being established intraoperatively. The possible severe complications that can put the patient’s life in danger are intestinal ischemia, torsion, strangulation, volvulus, and sepsis [4].

Madsen et al. suggested that the main cause of approximately half of cases is an error in the operative technique, such as breaking the suture through the fascia, widening the knot, weakening the suture, placing the knots at a distance, and the eventual protrusion of the abdominal contents (omentum and bowel) between the sutures [11]. After exploring the abdominopelvic viscera, the small bowel is carefully inspected, followed by its manual reduction and a corresponding parietorraphy. A therapeutic strategy regarding some recommendations for solving these conditions was developed by several international societies, particularly by the European Hernia Society [12].

The management of abdominal eviscerations has always represented a surgical challenge in terms of restoring the integrity of the abdominal wall. The surgical closure of the abdominal wall can be difficult because of the general condition of the patient, the increased risk of sepsis, possible damage to the abdominal contents, and the possible occurrence of postoperative complications (adhesions, fistulas, intra-abdominal hypertension, occlusions, and abdominal compartment syndrome). Closing the abdominal wall requires respecting the anatomy of the muscular–fascial layers, using specific surgical techniques, maintaining a good hydroelectrolytic and nutritional balance, and using appropriate suture materials [13].

We aimed to present a rare case of spontaneous abdominal evisceration after a TAH for a benign condition and to review the literature regarding these cases’ diagnosis and therapeutic management.

## 2. Case Presentation

A 68-year-old female presented to the emergency department 5 days post TAH with a bilateral salpingo-oophorectomy for uterine fibroids with complete abdominal evisceration. A distended abdomen was observed during the clinical examination due to abdominal tenderness, without signs of abdominal irritation, with dehiscence of the abdominal wound on the midline in the lower third, through which approximately 5 cm of the omentum was eviscerated and a small amount of serosanguineous secretion was externalized. Her medical history consisted of her last menstrual period 20 years ago, two vaginal births and no abortions, hypertension, chronic venous insufficiency, chronic thyroiditis, cholecystectomy, and appendicectomy. She tested negative for SARS-CoV2 at her admission to the hospital due to the pandemic context.

The indication for emergency surgical intervention was taken after the patient was stabilized in the intensive care unit. A total dehiscence of the fascia through which the omentum and intestinal loops were externalized was observed during the intraoperative inspection, classified as grade 1B according to the Björck classification (Table 1).

The intestinal loops were not affected, and a portion of the omentum, as well as the necrotic edges of the wound, was removed (Figure 1). Intraperitoneal cultures were taken, and a saline lavage was performed. Afterward, the surgical wound was closed without its edges in tension using an interrupted suture with a slowly absorbable monofilament 1–0 material with a suture-to-wound length (SL/WL) ratio of 4:1 (Figure 2 and Figure 3). The patient stayed in the intensive care unit for 48 h postoperatively, with the administration of antibiotics. The patient was discharged after 5 days in good condition. The 2-year follow-up showed no local signs of relapse.

## 3. Discussion

Midline laparotomy is one of the most common access routes into the abdominal cavity. This incision is used by general surgeons, vascular surgeons, gynecologists, and urologists. Midline incisions may extend from the xiphoid process to the pubic symphysis, depending on the organ upon which the surgical intervention will be performed and the operator’s needs. Midline laparotomies offer several advantages, as follows: they can be extended if greater exposure of the abdominal cavity is needed; they are easy to perform; they do not section essential vessels, muscles, or nerves; they require a short time for parietal recovery; and they heal via resistant scars [15].

We conducted an electronic search in the PubMed, Embase, and Web of Science databases because the available literature did not provide clear data regarding the pathogenesis and therapeutic management of the abdominal evisceration of the small intestine. We identified seven published cases with the following MeSH search terms: “abdominal wall dehiscence”, “abdominal evisceration”, “open abdomen”, “burst abdomen”, “abdominal fascial dehiscence”, “abdominal dehiscence post-hysterectomy”, and “hysterectomy complications”. The inclusion criteria were case reports, case series, and clinical trials published all-time in English and related to gynecological interventions.

The closure of abdominal wall incisions aims to restore its layers as anatomically as possible. The closure of the parietal peritoneum is usually performed with absorbable 2–0/3–0 sutures, and they are restored in approximately 6–7 h postoperatively. Several studies have not shown any benefits regarding the closure of the peritoneum [16,17]. Slowly absorbable or non-absorbable threads thicker than 1–0/2–0/0 are used for continuous sutures or separate threads of the aponeurosis. The aponeurosis is completely restored in approximately 3 months postoperatively. Quilting brings the anatomical structures closer for an easier skin suture with separate resorbable or non-resorbable threads. Skin recovery takes approximately 10–12 days postoperatively [15,18].

Wound healing occurs in two stages, as follows: one latent stage, which lasts approximately 4–6 days, and another fibroblastic proliferation stage, lasting for the next 6–8 days. Eviscerations can occur due to defects in the union of the abdominal walls or via disunion of the walls.

Four phases in the wound healing process have been identified that follow each other dynamically, continuously, and with clear mechanisms. Any interruption or alteration in the duration of the process can cause defects in the healing process. Macrophages play multiple roles in wound healing. In the early wound, a pro-inflammatory effect occurs initially via the release of cytokines, the activation of leukocytes, and the removal of apoptotic cells by macrophages. The first phase is hemostasis, followed by inflammation, proliferation, and remodeling. Each phase may be subject to the action of several factors (local and systemic) which can affect healing. The local factors act directly at the wound level, and the systemic ones refer to how the patient’s health is reflected in the healing capacity (Table 2). The inflammatory phase is characterized by the action of neutrophils, lymphocytes, and macrophages, with the release of cytokines and growth factors (TGF-β, FGF, and EGF) [19].

The next proliferative stage consists of tissue regeneration with the stimulation of keratinocytes, fibroblasts, angiogenesis, and the migration of T lymphocytes, respectively. CD4+ (T helper) cells intervene in healing alongside gamma–delta T cells of the skin (epidermal dendritic T cells) activated by keratinocytes, with the production of fibroblast growth factor 7 (FGF-7), keratinocyte growth factors, and insulin-like growth factor-1. Conversely, CD8+ cells (suppressor–cytotoxic T cells) inhibit wound healing [20].

Experimental studies have shown that the lack of T lymphocytes and gamma–delta keratinocytes delays wound closure. The proliferative phase is characterized by epithelial proliferation under the action of fibroblasts and endothelial cells with collagen generation, glycosaminoglycans, proteoglycans, and granulation tissue. The remodeling phase, with the newly formed capillaries’ regression and the wound’s contraction, occurs after the synthesis of the extracellular matrix [21].

Evisceration can have the following main causes: (a) preoperative causes, secondary to patient comorbidities, tissue quality (uremia, jaundice, and anemia), and an increased abdominal pressure (obesity, cough, constipation, ascites, and urinary retention); (b) operative causes related to the surgical technique (sutures that are too tight or too close to the edge and unsafe knots); and (c) postoperative complications associated with wound infection [3] (Table 3).

Increased morbidity and mortality rates are observed in high-risk patients for abdominal dehiscence with evisceration, with additional bracing required for abdominal wall closure [18]. The identified risk factors for abdominal evisceration are an age of > 65 years, male sex, healing defects, the type of incision, incorrect wound closure techniques, hemodynamic instability, an increased intra-abdominal pressure (constipation and cough), emergency surgery, wound or abdominal wall infection, pelvic surgery, ascites, hypoproteinemia, anemia, diabetes, obesity, and radiotherapy. Other possible favorable factors are corticosteroid therapy, malignant tumors, and early postoperative lifting effort [5,10,22].

The management of evisceration involves the establishment of an urgent surgical treatment that includes the following two main points: cleaning the wound and the peritoneum and restoring the abdominal wall. Although infection is a risk factor in the pathogenesis of fascial dehiscence, the spectrum of pathogens incriminated could not be established. However, Gram-positive germs, especially Enterococci, have been identified in patients with FD after abdominal surgery [23].

In a retrospective study of 18,120 abdominal procedures, Haddad and Macon observed that 70 patients (0.38%) had abdominal dehiscence or evisceration, with a mortality rate of 5.5% and obesity as the main risk factor [22]. In another retrospective study carried out on 12,622 patients undergoing laparotomy over 9 years, 12 eviscerations were identified in women with a ratio of approximately 1:4 compared with men, with an incidence of 0.9 per thousand patients. In total, 84.2% of the patients required emergency surgery. The diagnosis was peritonitis in 45.6% of patients and intestinal occlusion in 33.3%. Reintervention was needed in 21% of cases. The surgical repair technique for the abdominal wall consisted of simple closing sutures. Mesh was used in 10.5% of patients [18]. In our patient, the fascial defect was immediately repaired without mesh.

Analysis over 15 years of 98,484 hysterectomies identified that 74% had indications of benign disease and 26% of malignant conditions, with a decrease in the annual rate of hysterectomies from 12% in 2000 to 5.53% in 2015. Thus, the incidence rate of hysterectomy according to age was 351.1 per 100,000 person-years. The trend was of a transition from TAH to minimally invasive surgical procedures, which led to a decrease in the rate of postoperative complications [24].

There is scarce research to support any specific repair technique in this setting. The initial therapeutic conduct in these situations aims to stabilize the patient in intensive care, correct imbalances, and preoperative preparation to resolve the evisceration. Appreciation of the eviscerated structures is very important in creating the decision scheme. The abdominal contents must be handled carefully, respecting asepsis principles. The wound must be completely opened and cleaned quickly, followed by collecting samples for cultures. The eviscerated structures must be isolated with sterile dressings moistened with warm saline. After entering the peritoneal cavity, all pathological processes, such as abscesses, fistulas, or adhesions, must be resolved. The next step is to wash the peritoneal cavity with saline thoroughly. Areas of necrosis must be excised to healthy tissue. Wall restoration occurs in the anatomical layers. The tissues are very friable and infiltrated after the excision of the necrotic edges. Some wound edges may be in tension when threads are placed in a “U” or “S” formation. Postoperative treatment with vitamins, human plasma, and crystalloids is instituted. Antibiotic prophylaxis is practiced, and gastric aspiration is recommended until transit resumes. The diet must be high-protein, high-calorie, and vitamin-rich [15,18].

The EHS guidelines, good practice statements, and clinical expertise guidelines follow the methods used to close the muscular–fascial layers of the abdominal wall and the possible materials that can be used, using the GRADE (Grading of Recommendations Assessment, Development, and Evaluation) approach. The main recommendations in the case of an open abdomen (OA) are the options of clinical expertise regarding the method for closing the suture, the preparation of the components, the degree of reinforcement of the stitches, the dynamic closure of the abdominal wall, and the avoidance of relaxing incisions [12]. The methodological guidelines of SIGN (Scottish Intercollegiate Guidelines Network) regarding the decrease in the incidence of IHs recommend the use of selective approaches on the midline of the laparotomy, of continuous sutures, with monofilament threads with monolayer slow absorption, and without separate closure of the peritoneum. In elective laparotomies, fascial closure techniques with small bites and an SL/WL of at least 4:1 are indicated. These recommendations are indicative in emergencies and obese patients, preferring techniques with a prophylactic mesh enlargement [25].

In the case of the complete dehiscence of the abdominal wound, mass closure according to Jenkins’ rule is preferred (with the ideal ratio of SL/WL being 4:1) to prevent the additional tension of sutures that will cause the aponeurosis to be cut [26]. Non-closure of the peritoneum after TAH does not represent a risk factor for evisceration [27].

In a prospective study including 60 patients with a subcutaneous adipose tissue thickness of more than 2.5 cm, Kore et al. observed that the closure of subcutaneous tissue after TAH decreased the rate of the occurrence of seromas, hematomas, and other complications of the wound [28]. In a unique, retrospective case–control study, the degree of abdominal rectus diastasis was established via CT imaging evaluation as a minimum width of at least 3 cm measured at 3 cm above the umbilicus. Closure of the midline laparotomy aponeurosis used a continuous suture with slowly absorbable threads with 5 mm bites and a minimum SL/WL ratio of 4:1. The result of the study highlighted the association between rectal diastasis and a burst abdomen (BA) [29].

The CONTINT multicenter randomized controlled trial conducted by Polychronidis et al. showed no difference between continuous and interrupted abdominal wall closure in the case of BA after emergency midline laparotomy [30]. An analysis of 131 patients with relaparotomy and 50 with primary laparotomy highlighted prolonged access to the abdomen, a higher peritoneal adhesion index, and more frequent accidental enterotomies in the group with relaparotomy, without observing differences in the rate of dehiscence of the wound with evisceration [31]. Reducing the rate of fascial dehiscence requires the use of a standardized technique for the primary repair of the abdominal wall defect with mass sutures with slowly absorbable threads with a depth of 3 cm and a pitch of 5 mm, keeping an SL/WL of 10:1 [32].

Predicting the occurrence of evisceration in high-risk patients is very important, requiring the development of risk calculation models in recent years. The validation of a risk model on high-risk patients was associated with a high predictive value for abdominal wound dehiscence, with the score ranging from 0 to 8.5, with a major increase in the risk of abdominal wound dehiscence from 0.02% to 70.1% [4]. Another predictive calculation model for the risk of fascial dehiscence after exploratory laparotomy was based on a study group of 594 patients, in which it was detected in a percentage of 6.9%, where a score of over three corresponded to an 18% risk. This machine learning model had a sensitivity of 70% and a specificity of 80%. In addition, the following main risk factors were analyzed: chronic obstructive pulmonary disease, immunosuppression, smoking, anticoagulant administration, the infectious process, and obesity [33].

The main predictors of the risk model, analyzed in a retrospective cohort study performed on 93,024 patients undergoing exploratory laparotomy, were operative time, wound infections, and obesity, with the clinical benefit varying between 0.8% and 4.5% [34]. In another study including 504 patients, 111 of which had a BA, the following predictive factors for BA were observed: intestinal resections, liver cirrhosis, and emergency surgery. Using interrupted sutures and possibly implanting an absorbable mesh are recommended to prevent BA recurrence [35].

Thus, the standardization of surgical techniques for the primary repair of OA contributed to a decrease in the incidence of fascial dehiscence processes [32]. In a prospective, single-center, non-randomized, controlled cohort study in which 351 patients (143 women) were included, a net benefit was observed with the use of a short suture with an SL/WL of 6:1 in midline incisions with a slowly absorbable 2–0 suture material, which was achieved in only 43% of cases (85% > SL/WL of 4:1). The closure should be performed in two layers in the case of transverse incisions [36].

In a multicenter prospective study, 202 patients with Björck grade 1A (Table 1) primary abdominal wall dehiscence following midline laparotomies were treated using posterior component separation and muscle release transversus abdominis reinforced by retro-muscular mesh insertion. This procedure prevented recurrences of IH and reduced mortality to 2.5% [37].

Several studies have shown an 85% reduction in the rate of postoperative IH in high-risk patients using only mesh reinforcement compared with primary suture closure secondary to midline laparotomy, without changing the wound infection incidence [38,39,40]. Furthermore, the recent literature has suggested that using mesh in contaminated fields can lead to outcomes similar to those of non-mesh repair in the same setting [41].

In abdominal emergencies, negative pressure therapy determines the increase in the rate of direct fascial closure and the reduction in the occurrence of entero-atmospheric fistulas. Extending this procedure beyond a week leads to the formation of a frozen abdomen without reducing sepsis [42]. According to the guidelines, one can opt for the temporary closure of the wall in the therapeutic management of the OA. At this time, combined techniques of vacuum-assisted dynamic fascial closure and mesh-mediated traction are recommended [43] (Figure 4).

Evisceration prophylaxis is crucial and part of the therapeutic plan for adequately monitoring these patients. A therapeutic management plan is necessary in the event of an evisceration (Figure 5).

Postoperatively, evisceration is correlated with prolonged hospitalization, the need to institute intensive therapy maneuvers, an increased morbidity and mortality, and additional costs. Thus, identifying patients with associated risk factors, proper counseling, reducing operative technique errors via a process of the continuous medical training of young specialists, increasing the number of minimally invasive interventions, and resolving these cases in tertiary hospital units will contribute to better prevention.

New research directions are trying to validate techniques based on the implantation of tension sensors that analyze the information recorded in real time from the suture level to better understand the pathogenesis of IH and reduce the occurrence rate of evisceration [44].

Klink et al. recorded the suture tension for 60 min to describe the correlation between the collagen content of the sutured tissue and the reduction in suture tension. Thus, the loss of tension is lower in the case of an increased collagen content [45]. Experimental studies in pigs showed that the suture-level tension decreased by 26% after one hour. Hence, it decreased by 45% after approximately 23 h. This tension dynamic at the suture level is important, because 15–20% of patients present with either increased intra-abdominal pressure or abdominal compartment syndrome [46].

Surgical wound dehiscence negatively affects physical activity, delaying the social reintegration of patients and influencing their quality of life and mental health [47]. Thus, patients with severe abdominal injuries who underwent staged abdominal repair (STAR) had a lower quality of life than that observed in patients suffering from non-STAR trauma [48]. Another study regarding the improvement of quality of life, assessed by the EQ-5D questionnaire, was shown to be correlated with the size of the hernia and with certain patient-related factors (diabetes, cardiovascular disease, and age of > 60 years) [49]. In recent years, the introduction of artificial intelligence (AI) algorithms has generated a reduction in healing time, an increase in quality of life, and a decrease in complication rates [50].

## 4. Conclusions

Abdominal evisceration of the small-bowel loops after pelvic surgery is an uncommon complication that requires urgent surgical intervention. Because of its nonspecific symptomatology, preoperative counseling about the possible complications of hysterectomy is important for patients with risk factors. Postoperatively, fascial dehiscence requires early identification for repair by a multidisciplinary team due to morbidity and mortality, as well as the patient’s low quality of life.

## Figures and Tables

**Figure 1 jcm-14-00262-f001:**
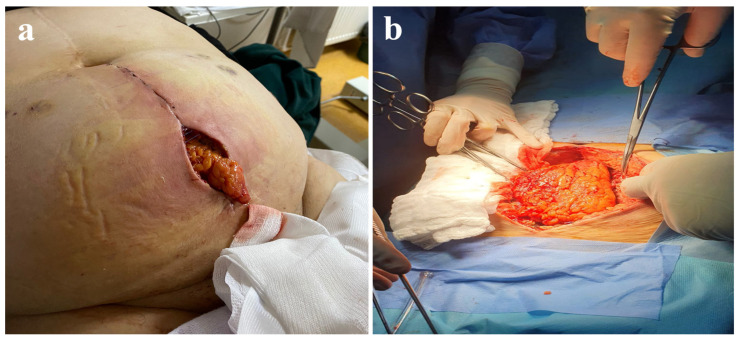
(**a**) Abdominal evisceration of the omentum and below the small-bowel loops. (**b**) Intraoperative image of the total fascial dehiscence after the prolonged skin incision.

**Figure 2 jcm-14-00262-f002:**
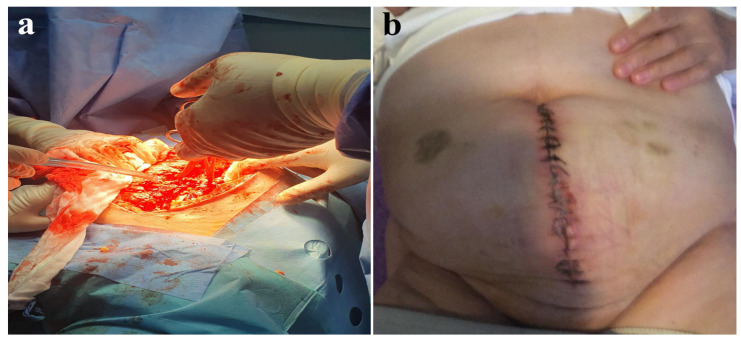
(**a**) Closure of the abdominal wall without tension. (**b**) Postoperative evaluation at one week.

**Figure 3 jcm-14-00262-f003:**
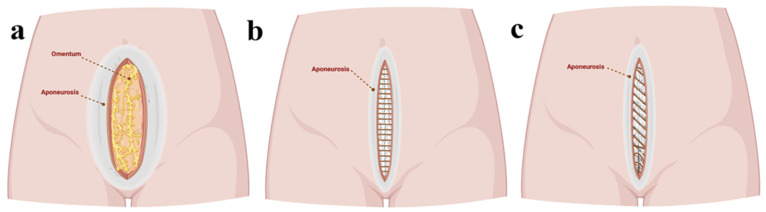
Schematic diagram of the case. (**a**) Complete dehiscence of the abdominal wall; (**b**) we prefer the interrupted sutures (SL/WL ratio <4:1) to repair the dehiscence of the anatomical planes of the abdominal wall; and (**c**) another possible scenario with continuous sutures.

**Figure 4 jcm-14-00262-f004:**
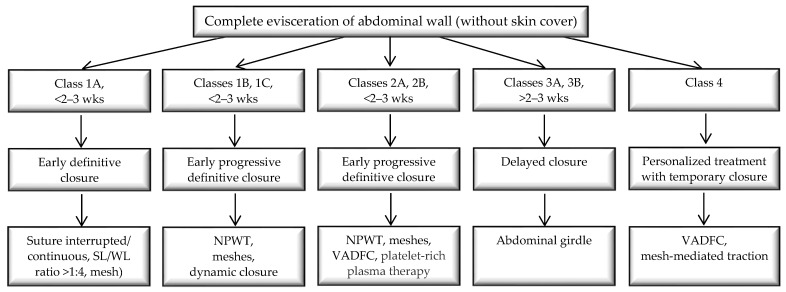
Flow diagram of management of complete abdominal evisceration according to Björck classification. (NPWT—negative pressure wound therapy; VADFC—vacuum-assisted dynamic fascial closure; wks—weeks.)

**Figure 5 jcm-14-00262-f005:**
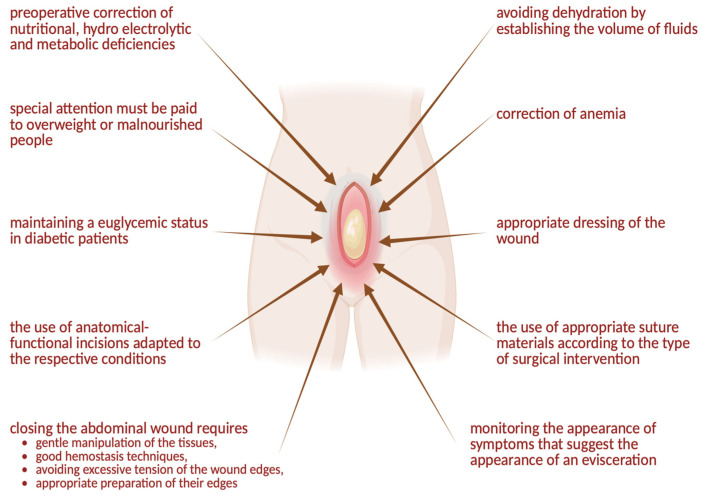
General recommendations regarding the prevention of evisceration.

**Table 1 jcm-14-00262-t001:** Classification of open abdomen according to Björck et al [14].

**1**	A	Clean OA, no adherence between bowel and abdominal wall or fixity
	B	Contaminated OA, no adherence/fixity
	C	Enteric leak, no fixation
**2**	A	Clean OA, plus adherence/fixity
	B	Contaminated OA, plus adherence/fixity
	C	Enteric leak, plus fixation
**3**	A	Clean, frozen abdomen
	B	Contaminated, frozen abdomen
**4**		Established entero-atmospheric fistula, frozen abdomen

**Table 2 jcm-14-00262-t002:** Systemic factors affecting wound healing and the risk of abdominal wall evisceration.

	Systemic Factors Impaired	Diabetes	Obesity	Advanced Age	Smoking/ Alcoholism	Drugs (Corticoids and Chemotherapics)	Stress	Nutrition
Wound Healing								
Impaired angiogenesis/neovascularization		x	x	x	x	x	-	-
Hypoxia		x	x	-	x	x	-	-
Fibroblast/keratinocyte dysfunction		x	-	-	-	x	-	x
Impaired immune response		x	x	-	-	-	x	x
Infection risk		x	x	-	x	-	-	-
Impaired vascular flow		-	x	-	-	-	-	-
Impaired wound contractions		-	-	-	-	x	-	-
Altered inflammatory response		-	-	x	x	-	x	-

**Table 3 jcm-14-00262-t003:** Perioperative risk assessment of abdominal evisceration.

Risk Factors		
Preoperative	Intraoperative	Postoperative
Old age Male sex High BMI Malignant tumors Anemia Smoking Radiotherapy Corticosteroid therapy Comorbidities - Asthma - Diabetes mellitus - Hypertension	Type of hysterectomy Type of incision Emergency surgery Long operation time Complexity of operation Incorrect wound closure techniques ASA score > 3 Amount of bleeding Hemodynamic instability Wound/abdominal wall infection Intraabdominal sepsis Hypoproteinemia Ascites	Erythrocyte suspension transfusion Albumin transfusion Antibiotics usage Early lifting effort

## Data Availability

Data are contained within the article.

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
