# Peer review of "Complete Abdominal Evisceration After Open Hysterectomy: A Case Report and Evidence-Based Review"

_jcm, 2025, doi:10.3390/jcm14010262_

Round 1
Reviewer 1 Report
Comments and Suggestions for Authors
Dear authors thank your efforts for addressing this rare but important problem.
Unfortunately, there are many English grammar problems in your manuscript, which should be corrected. Some of them
At the line 53-57, the sentence should be re-written, too difficult to understand.
At the line 58-59, the sentence has similar problem.
At the line 71-72, the sentence should be checked for English grammar.
At the line 73-74, the sentence is scientifically discussible. The most common reason of transvaginal evisceration should be MIS (laparoscopic hyterecyomy).
At the line 80-83, the sentences should be checked and be re-written.
Figure 3 is not necessary. It may remove out from manuscript.
At the line 152-155, the sentences should be checked and be re-written. Also there are many sentences, which should be checked for English grammar.
English language should totally be checked in this manuscript.
Comments on the Quality of English LanguageEnglish language should totally be checked in this manuscript.
This study totally should be re-written
Author Response
Dear authors thank your efforts for addressing this rare but important problem.
Unfortunately, there are many English grammar problems in your manuscript, which should be corrected. Some of them:
At the line 53-57, the sentence should be re-written, too difficult to understand.
Answer: Thank you very much for taking the time to review this manuscript. We rewrote the sentence. The corresponding revisions are in the new version of the manuscript, see lines 53-56.
At the line 58-59, the sentence has similar problem.
Answer: Thank you for your suggestion. We rewrote the sentence and moved it to the end of the introduction section; see lines 103-104.
At the line 71-72, the sentence should be checked for English grammar.
Answer: Thank you for your remark. We corrected the sentence. Please see lines 68-70.
At the line 73-74, the sentence is scientifically discussible. The most common reason of transvaginal evisceration should be MIS (laparoscopic hyterecyomy).
Answer: Thank you for pointing this out. We did not want to go into detail and discuss the reason for transvaginal evisceration in this sentence, but rather to point out that it is more commonly reported in the literature than abdominal evisceration, as this paper focuses on abdominal evisceration. Thus, we have rewritten the sentence for better understanding. (please see lines 70-74) Current published data show that the risk of vaginal cuff dehiscence after hysterectomy using minimally invasive surgical methods (laparoscopy and robot-assisted surgery) is not higher compared to abdominal hysterectomy, but is higher than after total vaginal hysterectomy. (doi: 10.1016/j.amsu.2022.103986; doi.org/10.3390/jcm12083001)
At the line 80-83, the sentences should be checked and be re-written.
Answer: Thank you for your mention. We rewrote the sentence. Please see the attached manuscript, lines 77-83.
Figure 3 is not necessary. It may remove out from manuscript.
Answer: Thank you for this mention. The authors agree that Figure 3 with a schematic diagram of the case is not necessary, and we have removed it from the paper. Please consult the attached manuscript.
At the line 152-155, the sentences should be checked and be re-written. Also there are many sentences, which should be checked for English grammar.
English language should totally be checked in this manuscript.
Answer: Thank you for your mention. We have modified the sentence according to your suggestion and revised the entire article regarding the grammar errors. Please see lines 150-153.
Kind regards
Reviewer 2 Report
Comments and Suggestions for Authors
Dear Author,
the manuscript itself is scientifically well written. I have only few suggestions that may improve its quality.
Please correct it in the places (wherever possible, especially in discussion section) pointed out in iThenticate report to diminish the rate of overlap.
Please bare in mind that epidemiology of evisceration is biased. Nobody wants to share this embarassing data, so writing that eviscaration is extremly rare is not true. Please correct introduction and discussion accordingly or put critical comments.
Author Response
Dear Author,
the manuscript itself is scientifically well written. I have only few suggestions that may improve its quality.
Please correct it in the places (wherever possible, especially in discussion section) pointed out in iThenticate report to diminish the rate of overlap.
Answer: Thank you very much for taking the time to review this manuscript and for appreciating our work. We have made the changes you suggested, following the iThenticate report to reduce overlap.
Please bare in mind that epidemiology of evisceration is biased. Nobody wants to share this embarassing data, so writing that evisceration is extremely rare is not true. Please correct introduction and discussion accordingly or put critical comments.
Answer: Thank you for pointing out that the epidemiology of evisceration is biased. We have clarified this aspect and corrected the data on the epidemiology of evisceration throughout the manuscript. Please see the attached manuscript.
Kind regards
Reviewer 3 Report
Comments and Suggestions for Authors
Abstract
Would you kindly define better your inclusion and exclusion criteria in your literature review?
Discussion
I think it would be interesting to provide more information from the literature regarding the association between wound healing phases and the risk factors leading to wound dehiscence?
Also, you might create a table where you systemize the risk factors for developing evisceration.
Furthermore, readers might be interested in learning more about the 'implantation of suture tension sensors'. Would you kindly offer more information regarding this subject?
Conclusion:
I suggest you expand this section. In addition, you might write more data about the quality of life of patients with evisceration in Discussion section (in order to support your Conclusion: 'as well as the patient's low quality of life').
References:
I have noticed the increased number of references in this paper. However, I suggest the authors include the ones published within the last ten years timeframe.
Comments on the Quality of English LanguageI think that using more English connectors might improve the aspect of the text, making it more comprehensive.
Author Response
Abstract
Would you kindly define better your inclusion and exclusion criteria in your literature review?
Answer: Thank you very much for taking the time to review this manuscript. We have completed the methodology with more detail on the study inclusion criteria. Please see the lines 154-161.
Discussion
I think it would be interesting to provide more information from the literature regarding the association between wound healing phases and the risk factors leading to wound dehiscence? Also, you might create a table where you systemize the risk factors for developing evisceration. Furthermore, readers might be interested in learning more about the 'implantation of suture tension sensors'. Would you kindly offer more information regarding this subject?
Answer: Thank you for suggesting this additional information. We have incorporated additional discussions about the association between wound healing phases and risk factors leading to wound dehiscence, about the "implantation of suture tension sensors," and created a table systematizing risk factors for the development of evisceration. This can be found in the resubmitted article at lines 175-198, 370-379, and in tables 1 and 2.
Conclusion:
I suggest you expand this section. In addition, you might write more data about the quality of life of patients with evisceration in Discussion section (in order to support your Conclusion: 'as well as the patient's low quality of life').
Answer: Thank you for the suggestion. We agree that more information should be included about patients' quality of life with evisceration and have updated the talk section. Please see lines 377-386.
References:
I have noticed the increased number of references in this paper. However, I suggest the authors include the ones published within the last ten years timeframe.
I think that using more English connectors might improve the aspect of the text, making it more comprehensive.
Answer: Thank you for the mention. While abdominal evisceration was the main focus of the paper and the literature reporting is limited, we have performed a permanent search in the databases. We have modified some references and sent the paper for proofreading in English. Please see the attached manuscript.
Kind regards
Round 2
Reviewer 3 Report
Comments and Suggestions for Authors
Thank you for taking into account my suggestions! I consider that this manuscript is suitable for being published, advancing the current knowledge in the field.
Best regards!